# Mesenchymal stem cell-neural progenitors are enriched in cell signaling molecules implicated in their therapeutic effect in multiple sclerosis

Violaine K. Harris ⬤*, Jaina Wollowitz ⬤[¤a], Jacelyn Greenwald ⬤[¤b], Alyssa L. Carlson ⬤[¤c], Saud A. Sadiq

Tisch Multiple Sclerosis Research Center of New York, New York, New York, United States of America

¤a Current address: Tri-Institutional PhD Program in Chemical Biology, Weill Cornell College of Medicine, New York, New York, United States of America
¤b Current address: Department of Chemistry and Biochemistry, Ohio State University, Columbus, Ohio, United States of America
¤c Current address: College of Medicine, State University of New York Downstate Health Sciences University, Brooklyn, New York, United States of America
* vharris@tischms.org

**Data Availability Statement:** All relevant data are within the paper and its Supporting Information files.

## Abstract

Mesenchymal stem cell-neural progenitors (MSC-NP) are a neural derivative of MSCs that are being investigated in clinical trials as an autologous intrathecal cell therapy to treat patients with secondary progressive (SP) or primary progressive (PP) multiple sclerosis (MS). MSC-NPs promote tissue repair through paracrine mechanisms, however which secreted factors mediate the therapeutic potential of MSC-NPs and how this cell population differs from MSCs remain poorly understood. The objective of this study was to define the transcriptional profile of MSCs and MSC-NPs from MS and non-MS donors to better characterize each cell population. MSCs derived from SPMS, PPMS, or non-MS bone marrow donors demonstrated minimal differential gene expression, despite differences in disease status. MSC-NPs from both MS and non-MS-donors exhibited significant differential gene expression compared to MSCs, with 2,156 and 1,467 genes upregulated and downregulated, respectively. Gene ontology analysis demonstrated pronounced downregulation of cell cycle genes in MSC-NPs compared to MSC consistent with reduced proliferation of MSC-NPs *in vitro*. In addition, MSC-NPs demonstrated significant enrichment of genes involved in cell signaling, cell communication, neuronal differentiation, chemotaxis, migration, and complement activation. These findings suggest that increased cell signaling and chemotactic capability of MSC-NPs may support their therapeutic potential in MS.

## Introduction

Multiple sclerosis (MS) is a chronic autoimmune disease of the central nervous system. Relapsing-remitting MS (RRMS) is characterized by relapses of neuroinflammation, demyelination,

**Funding:** The authors received no specific funding for this work.

**Competing interests:** I have read the journal's policy and the authors of this manuscript have the following competing interests: Authors Violaine Harris and Saud Sadiq are listed as inventors on US patent #US 8,642,331, which is related to the study presented in the manuscript. The patent is issued to the Tisch MS Research Center of New York and is considered a non-financial competing interest. This does not alter our adherence to PLOS ONE policies on sharing data or materials. Furthermore, Violaine Harris and Saud Sadiq have no other relevant declarations related to employment, consultancy, products in development, etc. The remaining authors declare that they have no known competing financial interests or personal relationships that could have appeared to influence the work reported in this paper.

and neurological dysfunction followed by variable periods of remission. Over time, many RRMS patients develop secondary progressive MS (SPMS), a progressive course of neurological disability due to ongoing neuroaxonal damage. Primary progressive MS (PPMS) represents a smaller subset of MS patients (15%) in which the disease is characterized by progression from onset. Mesenchymal stem cells (MSCs) have emerged as a promising therapeutic modality for neuroinflammatory and neurodegenerative diseases including MS due to their capacity for immunomodulation and tissue repair [1]. In experimental models of MS, the therapeutic properties of MSCs overwhelmingly correlate with a robust secretome capable of promoting beneficial macrophage/microglia polarization, inhibiting Th1 and Th17 responses, ameliorating blood-brain barrier dysfunction and reducing inflammatory cell infiltration [1,2]. Recent clinical trials in MS have demonstrated the safety and feasibility of intravenously and/or intrathecally administered autologous bone marrow-derived MSCs, with some trends in clinical efficacy warranting further confirmation in randomized controlled studies [3–6].

To optimize the safety and efficacy of CNS-targeted MSCs, neural derivatives of MSCs have been investigated as an intrathecally (IT) administered cell therapy for MS [7,8]. Specifically, MSC-derived neural progenitor cells (MSC-NPs) have been developed as a treatment strategy based on their neural characteristics, including neurosphere morphology and upregulation of neural genes. MSC-NPs also have a reduced capacity to differentiate into mesodermal lineages, which predicts enhanced safety due to reduced risk of ectopic differentiation when applied directly into the CNS [9–11]. Preclinical and clinical studies have demonstrated the therapeutic potential of MSC-NPs in MS. In the experimental autoimmune encephalomyelitis (EAE) mouse model of MS, IT injection of MSC-NPs administered after peak illness reduced clinical disease severity and decreased pathological evidence of neuroinflammation and demyelination [11]. A phase I clinical trial in twenty patients with progressive MS demonstrated that multiple intrathecal injections of autologous MSC-NPs were safe and well tolerated, with improved neurologic function in some patients including improved muscle strength and bladder function [8]. Subsequently, a 50-patient, randomized, placebo-controlled phase II clinical trial was initiated and is near completion.

As this novel cell therapy moves into clinical use, there is a need to further define and characterize MSC-NPs to better understand the mechanisms underlying their therapeutic efficacy and potency. This study aims to characterize the transcriptomic signature of MSC-NPs by identifying genes that were differentially expressed in MSC-NPs compared to donor-matched MSCs. Cells were derived from SPMS, PPMS, and non-MS donors to identify gene expression differences correlating with progressive MS. We found a robust shift in gene expression in MSC-NPs compared to MSCs that was independent of donor disease state and that was suggestive of increased cell signaling capacity in MSC-NPs that may be associated with therapeutic potential in MS.

## Materials and methods

### Ethics

Human bone marrow was obtained with ethics approval from Western IRB. Individuals who donated bone marrow provided written informed consent to participate in this study.

### Mesenchymal stem cell (MSC) and MSC-derived neural progenitor (MSC-NP) cell culture

Human MSCs were isolated from bone marrow as described previously [8]. MSCs were cultured in MSCBM basal media (Lonza) supplemented with 5% Plastate™ human platelet lysate

**Table 1. MSC donor demographics.**

| Donor ID/disease subtype | Age (years) | Gender | Disease duration (years) | Avg. PDT of MSCs (days) |
|---|---|---|---|---|
| CONT01 | 53 | M | n/a | 2.1 |
| CONT02[a] | 61 | F | n/a | 1.6 |
| SPMS03 | 68 | F | 12 | 1.6 |
| SPMS04 | 57 | F | 14 | 2.2 |
| SPMS05 | 53 | F | 21 | 1.3 |
| PPMS06 | 50 | M | 5 | 1.9 |
| PPMS07 | 51 | F | 20 | 1.4 |
| PPMS08 | 53 | F | 22 | 2.0 |

[a]CONT02 was a non-MS control from a stroke patient.

PDT, population doubling time; CONT, control; SPMS, secondary progressive multiple sclerosis; PPMS, primary progressive multiple sclerosis.

(New York Blood Center), 2 U/ml Heparin, and 2 mM GlutaMAX™I CTS™ to make MSC growth media (MSCGM). MSC-NPs were generated by culturing MSCs in neural progenitor maintenance medium (NPMM) (Lonza) supplemented with 20 ng/ml each of epidermal growth factor (EGF) and basic fibroblast growth factor (bFGF) for 2 weeks. Cells were incubated in a humidified 37°C incubator at 5% $CO_2$ and 5% $O_2$. All MSCs and MSC-NPs were derived from bone marrow from donors with multiple sclerosis participating in clinical trials (cliniclatrials.gov ID NCT03355365 and NCT03822858). Control (non-MS) MSCs were obtained from a stroke patient who was granted a single-patient expanded access IND to receive intrathecal MSC-NP treatment. Healthy control MSCs were obtained from whole fresh bone marrow purchased from AllCells. All MSCs and MSC-NPs were manufactured and quality tested in compliance with cGMP manufacturing [8]. A panel of 8 cell lines including 2 controls, 3 SPMS, and 3 PPMS was selected for RNA sequencing (Table 1). A separate panel of 2 healthy controls, 4 SPMS, and 4 PPMS was used for validation studies. Population doubling time (PDT) of MSCs was determined by the formula PDT = (t*log2)/(logN$_2$-logN$_1$) where t = number of days in culture, $N_1$ = number of cells seeded and $N_2$ = number of cells harvested.

## RNA sequencing

Total RNA for RNA sequencing was isolated from cells by phase separation using TRI Reagent (Sigma) followed by purification of ethanol-precipitated nucleic acid using RNeasy Mini spin columns (Qiagen). RNA quality and quantity were determined using Agilent RNA ScreenTape assay in combination with the TapeStation system. Bulk RNA sequencing was performed by Genewiz. Briefly, RNA libraries were prepared by poly-A selection and whole-transcriptome analysis was performed using Illumina HiSeq platform. RNA-Seq data were analyzed using Basepair software (https://www.basepairtech.com/) with a pipeline that included the following steps: 1) reads were aligned to the transcriptome derived from UCSC genome assembly hg19 using STAR with default parameters, and 2) read counts for each transcript were measured using feature counts. DESeq2 was used to compare differentially expressed genes (DEGs) between defined groups of samples. The Wald test was used to generate $p$-values and log2 fold changes. Genes with an adjusted $p$-value $< 0.05$ and absolute log2 fold change $> 1$ were identified as differentially expressed genes for each comparison.

Gene ontology (GO) enrichment analysis was performed using *GOrilla*, a publicly available GO enrichment tool to assess significance of enrichment for previously annotated and defined processes (GO terms) [12]. Enriched GO terms were identified using the list of target genes

"n" comprised of DEGs that were either upregulated (n = 2156) or downregulated (n = 1467) in the MSC-NP group compared to the MSC group. The background set of genes ("N") consisted of all 24,196 genes detected by RNAseq, of which 17,705 genes were associated with a GO term. "B" is the total number of genes associated with each specific GO term, and "b" is the number of genes in the intersection. Enrichment factor was determined by the formula (b/n) / (B/N). 'P-value' is the enrichment p-value computed according to the mHG (minimum hypergeometric) model and is not corrected for multiple testing of 15,354 GO terms. 'FDR q-value' is the correction of the p-value for multiple testing using the Benjamini and Hochberg method. FDR q-value cutoff was <0.05. For heatmaps, a Z-score normalization was performed on the normalized read counts across samples for each gene. Z-scores were computed on a gene-by-gene (row-by-row) basis by subtracting the mean and then dividing by the standard deviation. The computed Z score was then used to plot the heatmap.

## Quantitative PCR

Cells used for PCR validation were lysed in RLT Buffer with 1% 2-Mercaptoethanol (Qiagen). Total RNA was reverse transcribed using Superscript VILO Master Mix (Invitrogen). Quantitative real-time PCR (qRT-PCR) was performed using TaqMan Universal PCR Master Mix with pre-validated TaqMan gene expression assays (Thermo Fisher). TaqMan assays were *BST1* (Hs01070189_m1), *TGM2* (Hs01096681_m1), *MEST* (Hs00853380_g1), *CXCR4* (Hs00607978_s1), *NES* (Hs04187831_g1), *TLR2* (Hs00610101_m1), *HGF* (Hs00300159_m1), *SERPINF1* (Hs01106937_m1), *CSF1* (Hs00174164_m1), *CCL2* (Hs00234140_m1), *CCL5* (Hs99999048_m1), *MMP14* (Hs01037003_g1), *TNFAIP6* (Hs00200180_m1), *TIMP1* (Hs01092512_g1), *SPP1* (Hs00959010_m1), *C3* (Hs00163811_m1), *C7* (Hs00940408_m1), *CFI* (Hs00989715_m1), *C1R* (Hs00354278_m1), *C3AR1* (Hs00269693_s1), *A2M* (Hs00929971_m1) and endogenous control *IPO8* (Hs00183533_m1). PCR was performed using QuantStudio 7 Flex real-time qPCR (Thermo Fisher). Relative changes in gene expression were determined by ΔΔCt method with QuantStudio 7 Flex software (Thermo Fisher).

## BrdU incorporation assay

Cell proliferation was determined by BrdU Cell Proliferation ELISA Kit (Abcam). MSCs and MSC-NPs were plated in MSCGM and NPMM, respectively, at a concentration of 20,000 cells per well in a 96-well plate pre-coated with Matrigel. After 24 hours, media was changed into specified media with or without BrdU reagent. After 24 hours, cells were fixed and BrdU incorporation was measured according to manufacturer's instructions.

## Secreted protein analysis

Matched MSCs and MSC-NPs from the RNA-seq panel were passaged with TrypLE Express (ThermoFisher) and plated in Matrigel-coated wells at an equal density (range was 15,000–30,000 cells/cm$^2$) in MSCGM or NPMM, respectively. Conditioned media was collected after 48 hours, centrifuged for 10 minutes at 800 x g and 4˚C to remove cell debris, and stored at -80˚C before use. Secreted protein concentrations in conditioned media samples were determined by ELISA which included SerpinF1/PEDF (dilution 1:5), C3 (dilution 1:4000) (both from Abcam), HGF (undiluted), TIMP-1 (dilution 1:1000) and Osteopontin (dilution 1:5) (all from R&D Systems).

## Statistics

Data with multiple groups were analyzed by one-way ANOVA followed by Tukey's multiple comparison post-test. Differences in gene expression and protein levels between MSCs and

MSC-NPs were determined by paired t test. Correlations were determined by simple linear regression analysis. Statistical significance was set to a *p* value of < 0.05. Graphpad Prism 9 was used to calculate significance.

## Results

### MSC similarity between MS and non-MS donors

Bone marrow MSCs were isolated from 3 SPMS donors, 3 PPMS donors and 2 donors without MS (one healthy donor and one stroke patient). All donors were matched in age, gender, and disease duration (for MS donors), and isolated MSCs exhibited similar population doubling times (Table 1). MSCs and MSC-NPs were generated from each donor under identical conditions, and RNA-seq analysis was performed to identify differentially expressed genes in each group.

To determine whether MS disease subtype impacts gene expression in MSCs, gene expression in non-MS donors' MSCs (MSC-C) was compared to MSCs from SPMS donors (MSC-SP) and PPMS donors (MSC-PP). PCA analysis (Fig 1) showed minimal variance between MSCs across groups. Overall, few genes were differentially regulated in MSC-SP (116 genes) or MSC-PP (87 genes) relative to MSC-C, and only 17 of those genes overlapped between MSC-SP and MSC-PP groups (Fig 1). Gene ontology analysis did not reveal any significantly enriched pathways in any of the MSC groups. We selected several genes that shared similar differential expression in MS-derived MSCs compared to non-MS MSCs. Selected genes were validated by quantitative PCR utilizing a panel of RNA from both the RNA-seq panel and a separate validation panel of MSCs (Fig 1). We found that bone marrow stromal antigen 1 (*BST1)* was significantly upregulated in both the MS-derived MSC groups compared

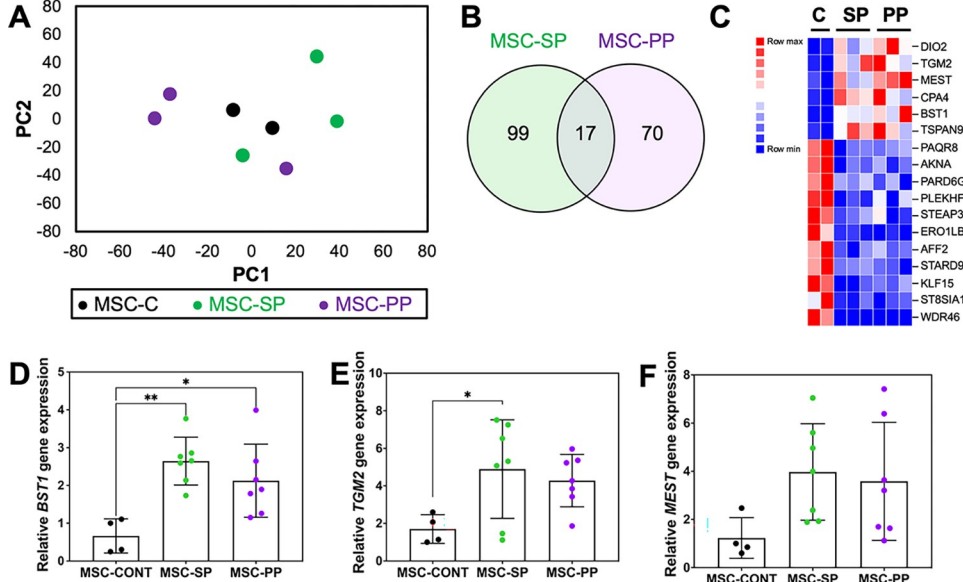

**Fig 1. High similarity between MSCs from SPMS, PPMS, and non-MS groups.** (A) Principal component analysis (PCA) of MSCs from non-MS (MSC-C), SPMS (MSC-SP), and PPMS (MSC-PP). MSCs cluster together independent of donor disease status. (B) Venn diagram showing the number of differentially expressed genes (DEGs) in MSC-SP and MSC-PP relative to control MSCs. Seventeen genes were differentially regulated in MS MSCs compared to controls. (C) Heatmap identifying 6 up-regulated and 11 down-regulated genes in MSC-SP and MSC-PP compared to MSC-C. (D-F) PCR validation of (D) BST1, (E) TGM2, and (F) MEST genes. Values represent mean ± standard deviation. *, p<0.05; **, p<0.01.

to the non-MS MSC group. (Fig 1). Transglutaminase-2 (*TGM2*) was significantly upregulated only in the MSC-SP group, and mesoderm-specific transcript (*MEST*) was upregulated in both MS groups but not reach significance compared to the non-MS group (Fig 1).

Similarly, very few DEGs were identified when comparing MSC-NPs from each donor population suggesting similarity of MSC-NPs across disease state. In MSC-NPs from SPMS donors (MSCNP-SP) and from PPMS donors (MSCNP-PP), 295 DEGs and 67 DEGs were identified, respectively, compared to the MSCNP-C group. Only 1 gene, *WDR46*, was downregulated in both MSCNP-SP and MSCNP-PP groups compared to the MSCNP-C group. Interestingly, *WDR46* was also downregulated in both MS-derived MSC populations compared to control MSCs (Fig 1). These results demonstrate that cells from MS and non-MS donors maintain a high similarity in gene expression, suggesting that MS disease status does not appreciably influence the transcriptomic profile of MSCs or their derived MSC-NPs.

## Shift in transcriptomic signature and cell cycle gene expression in MSC-NPs compared to MSCs

Due to the high degree of similarity between cell populations derived from MS and non-MS donors, we grouped all donor types for further analysis of MSC-NPs compared to MSCs. PCA analysis demonstrated that MSC-NPs clustered independently of MSCs, implying two distinct cell populations (Fig 2). The transcriptomic signature of MSC-NPs shifted dramatically, with 2,156 genes upregulated and 1,467 genes downregulated in MSC-NPs compared to MSCs (Fig 2).

Gene ontology analysis of the 1,467 genes downregulated in MSC-NPs compared to MSCs demonstrated an enrichment of cell cycle genes in MSC. Table 2 lists 10 representative GO pathways selected from a list of over 300 GO pathways (S1 Table) that were significantly enriched in MSCs. The majority of MSC-enriched GO terms were related to the regulation of cell cycle and proliferation, with additional pathways regulating angiogenesis, extracellular matrix reorganization, and response to hypoxia. Out of 608 genes in the mitotic cell cycle process pathway (GO:1903047), 133 genes were downregulated in MSC-NPs compared to MSCs (Fig 3 and S2 Table). The shift in cell cycle gene expression is consistent with differences in the culture conditions of MSCs compared to MSC-NPs. MSCs are highly proliferative as a result

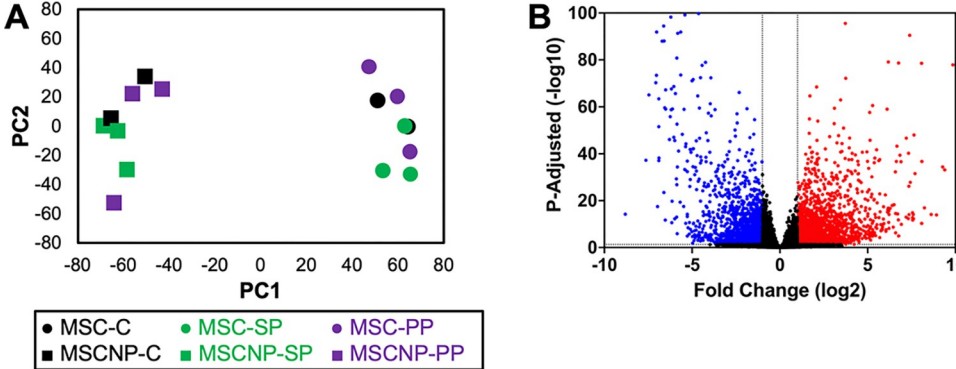

**Fig 2. Large shift in gene expression in MSC-NPs compared to MSCs.** (A) Principal component analysis (PCA) of MSCs (circles) and MSC-NPs (squares) from non-MS (MSC-C), SPMS (MSC-SP) and PPMS (MSC-PP) donors. MSCs and MSC-NPs cluster independently. (B) Volcano plot demonstrating upregulation of 2156 DEGs (red) and downregulation of 1467 DEGs (blue) in MSC-NPs compared to MSCs. Cells from C, SP, and PP donors were grouped together for DEG analysis. Dotted line indicates cutoff values of adjusted *p*-value < 0.05 and absolute log2 fold change > 1.

**Table 2. Selected GO terms enriched in MSCs compared to MSC-NPs.**

| GO Pathway Description | -Log10(FDR) | Enrichment Factor |
|---|---|---|
| Mitotic cell cycle process | 28.8 | 3.1 |
| Cell division | 18.1 | 3.2 |
| Cytoskeleton organization | 16.2 | 2.3 |
| Chromosome segregation | 14.3 | 5.2 |
| DNA replication | 8.5 | 3.6 |
| Microtubule cytoskeleton organization involved in mitosis | 7.4 | 3.8 |
| Extracellular matrix organization | 5.0 | 2.2 |
| Positive regulation of vasculature development | 4.5 | 2.5 |
| Response to hypoxia | 4.4 | 2.3 |
| Angiogenesis | 4.2 | 2.3 |

GO, gene ontology; MSC, mesenchymal stem cell; MSC-NP, MSC-derived neural progenitor; FDR, false discovery rate.

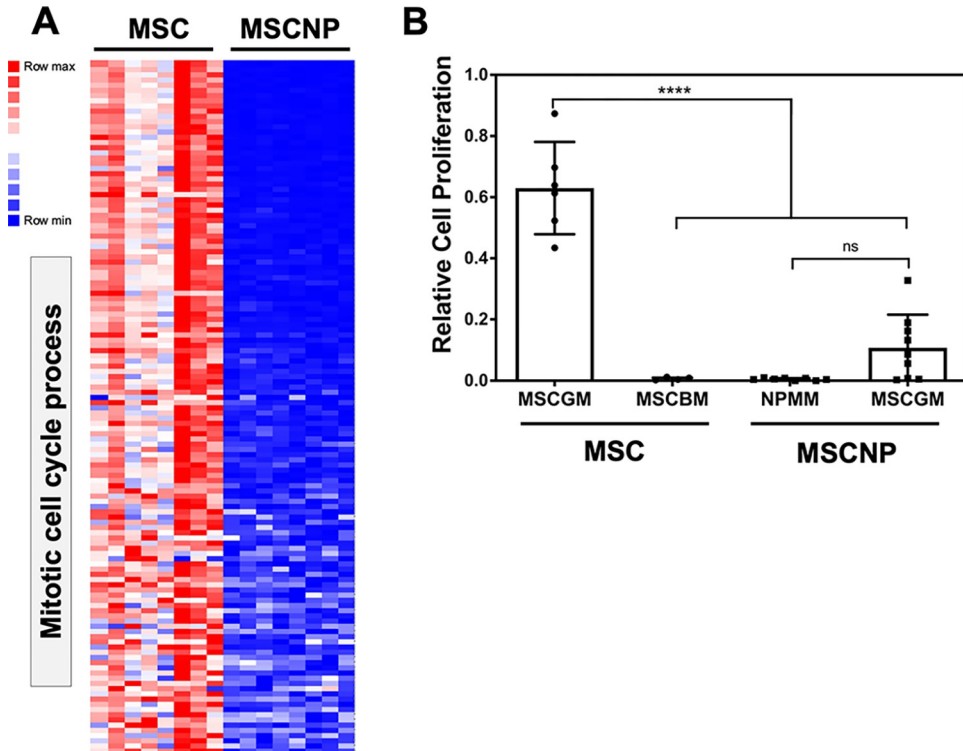

**Fig 3. Downregulation of cell cycle genes and reduced proliferation of MSC-NPs compared to MSCs.** (A) Heatmap representing downregulation of cell cycle genes in MSC-NPs compared to MSCs. All 133 overlapping genes in mitotic cell cycle process GO pathway (GO term GO:1903047) are shown. (B) BrdU incorporation assay demonstrating proliferative capacity of MSCs compared to lack of proliferation in MSC-NPs. Proliferation of MSCs in growth media (MSCGM) was arrested when cultured in basal media without growth supplement (MSCBM) for 24 hours. MSC-NPs do not proliferate under normal conditions (NPMM) and showed very limited proliferative capacity when cultured in growth media (MSCGM) for 24 hours. Values represent mean ± standard deviation. ****, p<0.0001.

of culturing in growth-promoting media containing serum or serum alternatives (MSCGM), and removal of serum results in cell cycle arrest as measured by BrdU incorporation (Fig 3). In contrast, MSC-NPs are cultured in the absence of serum (NPMM) and do not proliferate (Fig 3). When cultured in serum-containing MSCGM, MSC-NPs showed a variable and non-significant proliferative response (Fig 3), suggesting that MSC-NPs have fully exited the cell cycle.

## Neural genes upregulated in MSC-NPs

We next examined which gene pathways were enriched in MSC-NPs compared to MSCs. Gene ontology analysis of the 2,156 genes upregulated in MSC-NPs compared to MSCs identified multiple pathways as shown in Table 3, which lists 20 representative GO pathways selected from a list of over 300 GO pathways (S3 Table) that were significantly enriched in MSC-NPs. MSC-NP-enriched GO terms were related to cell signaling and communication, neuronal differentiation, chemotaxis and migration, and complement activation (Table 3). Out of 871 genes in the regulation of nervous system development pathway (GO:0051960), 165 genes were upregulated in MSC-NPs compared to MSCs (Fig 4 and S4 Table). Other similar pathways included regulation of neurogenesis, regulation of neuron differentiation, synaptic signaling, and neuron remodeling (Table 3). The enrichment of nervous system pathways in MSC-NPs supports their identity as a subpopulation of MSCs that exhibit neurosphere morphology and upregulation of neural genes [10]. Indeed, the panel of neural genes used for identity testing of MSC-NP batches during clinical manufacturing, specifically C-X-C motif chemokine receptor 4 (*CXCR4*), nestin (*NES*), toll-like receptor 2 (*TLR2*), and hepatocyte

**Table 3. Selected GO terms enriched in MSC-NPs compared to MSCs.**

| GO Pathway Description | -Log10(FDR) | Enrichment Factor |
|---|---|---|
| Cell-cell adhesion | 20.4 | 2.6 |
| Signaling | 17.1 | 2.3 |
| Cell-cell signaling | 16.2 | 2.3 |
| Cell communication | 14.5 | 2.0 |
| Regulation of nervous system development | 12.3 | 1.9 |
| Extracellular matrix organization | 11.8 | 2.5 |
| Inflammatory response | 11.8 | 2.3 |
| Chemotaxis | 10.2 | 2.4 |
| Regulation of leukocyte migration | 10.1 | 2.8 |
| Regulation of neurogenesis | 9.5 | 1.8 |
| Regulation of neuron differentiation | 7.9 | 1.8 |
| Synaptic signaling | 7.1 | 2.2 |
| Regulation of ossification | 6.0 | 2.4 |
| Regulation of leukocyte chemotaxis | 5.9 | 2.9 |
| Regulation of macrophage migration | 2.3 | 3.3 |
| Positive regulation of oligodendrocyte differentiation | 2.1 | 4.0 |
| Regulation of complement activation | 1.9 | 2.9 |
| Glial cell migration | 1.9 | 4.1 |
| Regulation of humoral immune response | 1.8 | 2.5 |
| Neuron remodeling | 1.7 | 4.9 |

GO, gene ontology; MSC, mesenchymal stem cell; MSC-NP, MSC-derived neural progenitor; FDR, false discovery rate.

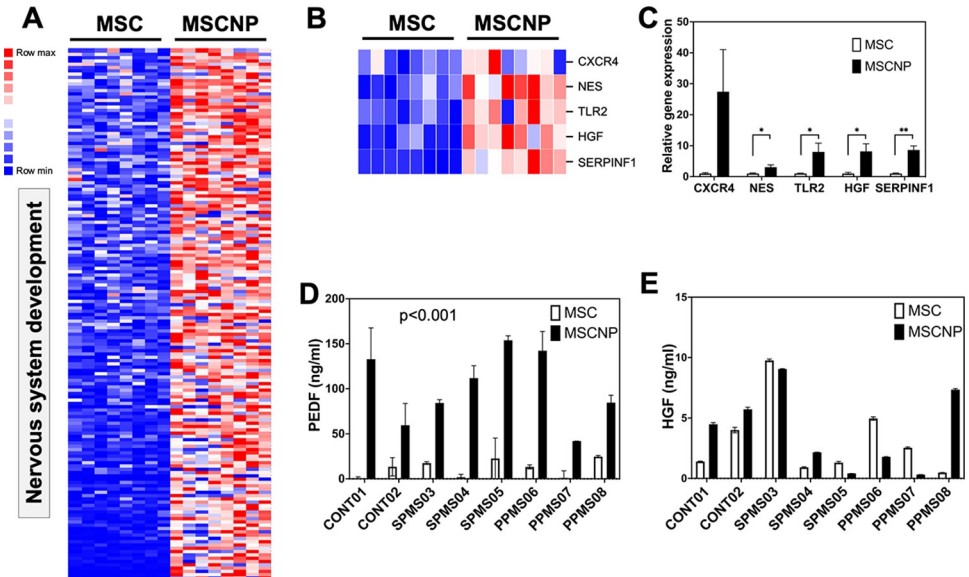

**Fig 4. Upregulation of neural genes verifies identity testing of MSC-NPs.** (A) Heatmap representing upregulation of neural genes in MSC-NPs compared to MSCs. All 165 overlapping genes in the regulation of nervous system development GO pathway (GO term GO:0051960) are shown. (B) Heatmap of 5 genes used in identity testing of MSC-NPs. (C) PCR verification of neural gene upregulation in MSC-NPs compared to MSCs. Values represent mean ± standard error. *, p<0.05; **, p<0.01 (D, E) Secreted protein concentration of (D) PEDF and (E) HGF measured by ELISA in MSC and MSC-NP supernatants. Mean differences across MSCs and MSC-NPs were statistically significant for PEDF (p<0.001) but not for HGF.

growth factor (*HGF*) was confirmed to be upregulated by both RNA sequencing and PCR (Fig 4). In addition, MSC-NPs were highly enriched for serpin family F member 1 (SERPINF1) gene expression (Fig 4). *SERPINF1* encodes the neuroprotective factor PEDF, which was present at a significantly higher concentration in MSC-NP conditioned media compared to MSC (Fig 4). Surprisingly, protein levels of HGF were elevated in only half of the MSC-NP conditioned media samples tested compared to MSCs, despite higher *HGF* gene expression in MSC-NPs (Fig 4). Overall, these findings support previous neural identification of MSC-NPs and reveal novel identity markers with potential mechanistic importance [10].

## MSC-NPs are enriched for genes regulating cell signaling

GO analysis demonstrated MSC-NPs were enriched for pathways involved in cell communication, including regulation of cell-cell signaling (GO:0007267), where 126 genes out of 530 were upregulated in MSC-NPs compared to MSCs (Fig 5 and S5 Table). In addition, MSC-NPs were enriched for genes involved in macrophage and glial cell migration as well as regulation of oligodendrocyte differentiation (Fig 5). Many of the genes in these categories encode secreted proteins that may be involved in the paracrine mechanisms underlying MSC-NPs therapeutic action, including C-X3-C motif chemokine ligand 1 (*CX3CL1*), C-X-C motif chemokine ligand 10 (*CXCL10*) and *HGF* which had been validated previously [10]. Utilizing an independent panel of MSC/MSC-NP pairs, the expression of additional secreted factors was validated by PCR and demonstrated significant upregulation of colony stimulating factor 1 (*CSF1*), C-C motif chemokine ligand 2 (*CCL2*), and matrix metalloproteinase 14 (*MMP14*), as well as variable upregulation of C-C motif chemokine ligand 5 (*CCL5*) (Fig 5). Expression of the tumor necrosis factor-inducible gene 6 (*TNFAIP6*) gene that encodes TSG-6 was tested

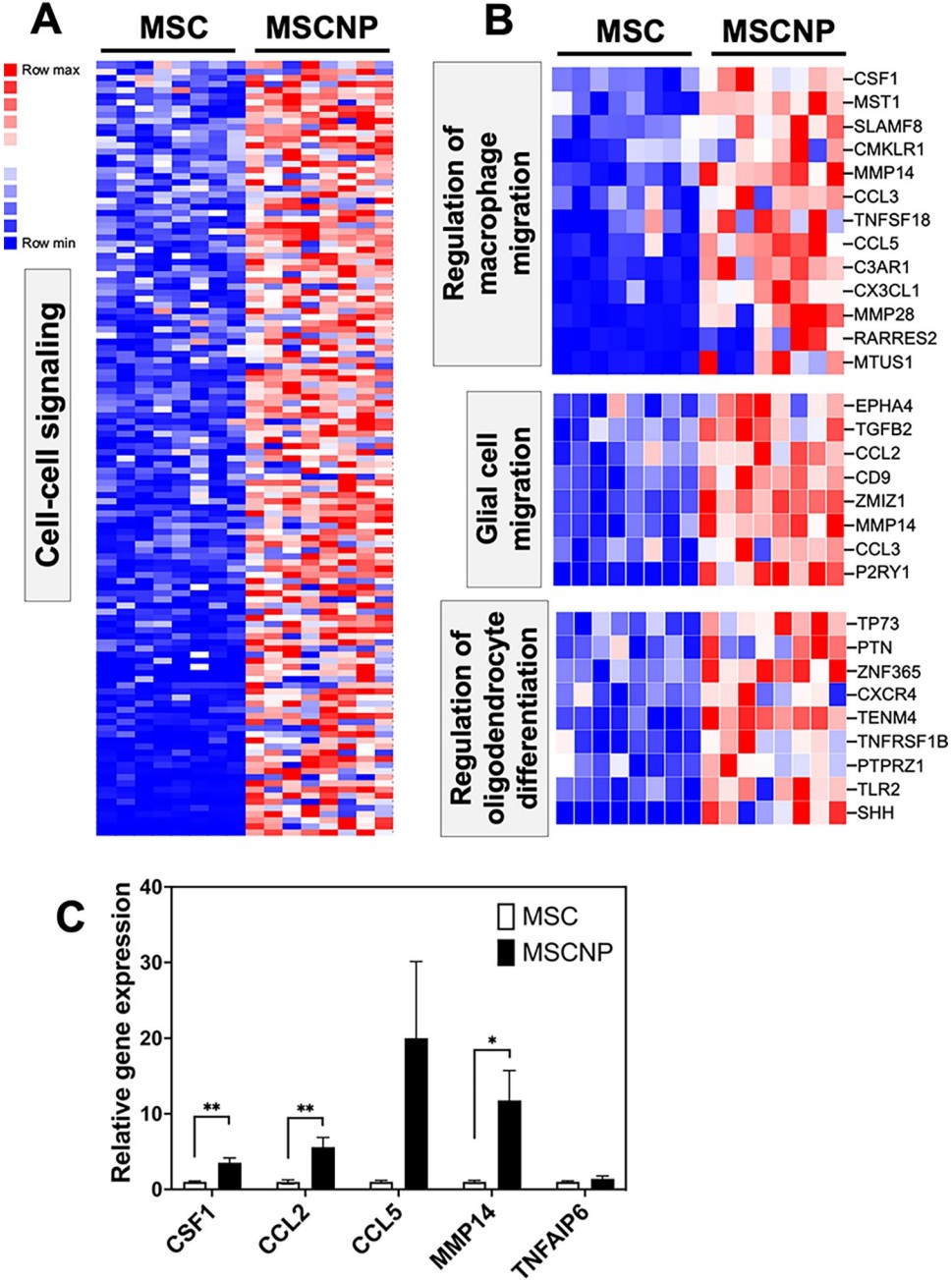

**Fig 5. Enrichment of cell signaling molecules in MSC-NPs (A) Heatmap representing upregulation of cell signaling genes in MSC-NPs compared to MSCs.** All 126 overlapping genes in the cell to cell signaling GO pathway (GO term GO:0007267) are shown. (B) Heatmaps depicting enrichment of genes regulating macrophage migration (GO term GO:1905521), glial cell migration (GO term GO:0008347), and positive regulation of oligodendrocyte differentiation (GO term GO:0048714) in MSC-NPs. (C) PCR validation of genes encoding secreted factors that are upregulated in MSC-NPs compared to MSCs. Values represent mean ± standard deviation. *, p<0.05; **, p<0.01.

due to its role in MSC-associated macrophage polarization, but was found not to be significantly upregulated (Fig 5) [13]. Overall, these findings suggest MSC-NPs are enriched for signaling molecules that may mediate the proposed paracrine mechanisms that underlie MSC-NP therapeutic activity.

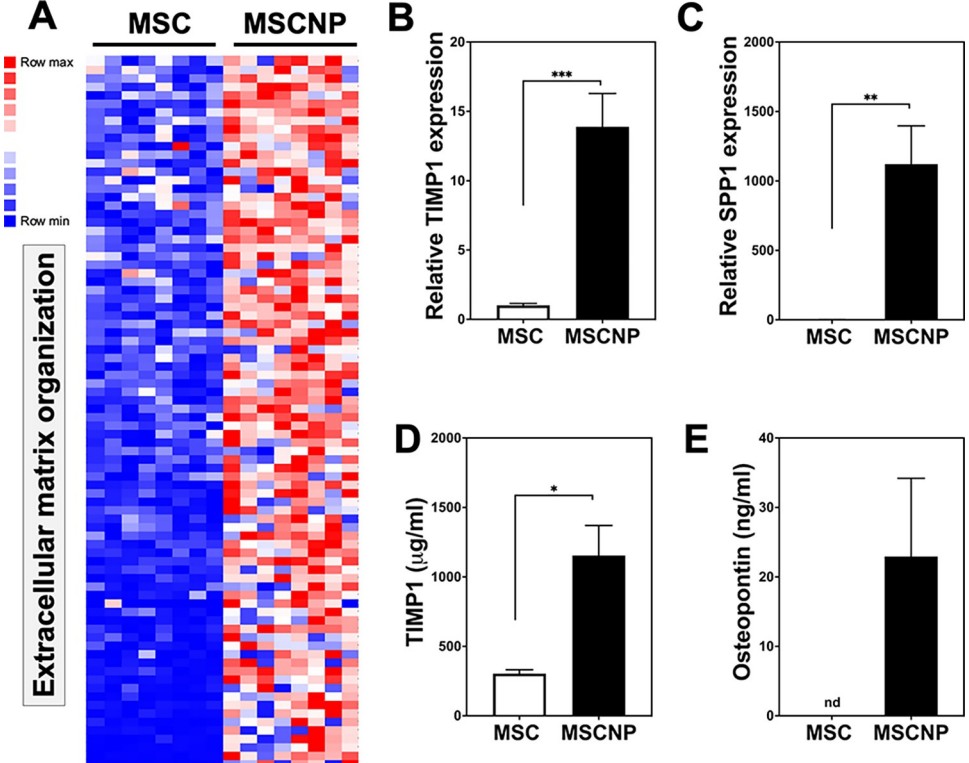

**Fig 6. Upregulation of genes involved in extracellular matrix organization in MSC-NPs.** (A) Heatmap representing upregulation of extracellular matrix organization genes in MSC-NPs compared to MSCs. All 84 overlapping genes in the extracellular matrix organization GO pathway (GO term GO:0030198) are shown. (B, C) PCR validation of (B) TIMP1 and (C) SPP1 gene upregulation in MSC-NPs compared to matched MSCs (n = 11 MSC/MSC-NP pairs). (D, E) Increased concentration of secreted (D) TIMP1 and (E) Osteopontin (protein encoded by SPP1) in conditioned media from MSC-NPs compared to MSCs as determined by ELISA. Osteopontin was not detected (nd) in conditioned media from MSCs. Values represent mean ± standard deviation. *, $p < 0.05$; **, $p < 0.01$; ***, $p < 0.001$.

## MSC-NPs enriched for genes regulating extracellular matrix organization

Extracellular matrix (ECM) production and remodeling likely play an important role in the homeostatic and regenerative functions of MSCs [14]. Consistent with this finding, we observed that genes with the highest normalized read counts in both MSCs and MSC-NPs included ECM genes encoding collagen subtypes (*COL1A1*, *COL1A2*, *COL3A1*, *COL4A1*, *COL4A2*, *COL5A1*, *COL6A1*, *COL6A2*, *COL6A3*), fibronectin (*FN1*), fibrillin (*FBN1*), thrombospondin (*THBS1*, *THBS2*), galectin (*LGALS1*) and integrin (*ITGA5*, *ITGB1*), as well as genes encoding ECM remodeling enzymes including MMPs (*MMP2*, *MMP14*) and TIMPs (*TIMP1*, *TIMP2*, *TIMP3*). Interestingly, GO analysis demonstrated that MSC-NPs were enriched for pathways that regulate ECM organization (GO:0030198), where 84 out of 336 pathway genes were enriched in MSC-NPs compared to MSCs (Fig 6 and S6 Table). Enriched genes encoded several secreted factors including TIMP1 and SPP, which were confirmed to be significantly upregulated in MSC-NPs compared to MSCs (Fig 6). In addition, significantly higher levels of secreted TIMP1 and Osteopontin (protein encoded by SPP1) were detectable in MSC-NPs (Fig 6).

## MSC-NPs enriched for genes involved in humoral immune response

Finally, gene enrichment analysis demonstrated that MSC-NPs have increased expression of genes involved in the humoral immune response (Table 3) and complement activation

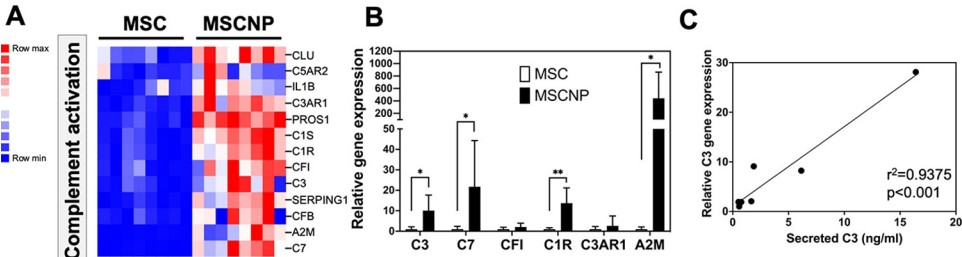

**Fig 7. Upregulation of complement pathway in MSC-NPs.** (A) Heatmap representing enrichment of complement genes in MSC-NPs compared to MSCs. All 13 overlapping genes in the regulation of complement activation GO pathway (GO term GO:0030449) are shown. (B) Validation of complement gene upregulation in MSC-NPs compared to MSCs by quantitative PCR. Values represent mean ± standard deviation. *, p<0.05; **, p<0.01 (C) Correlation between C3 secretion and C3 gene expression (normalized read counts) in MSC-NPs (MSC-derived C3 could not be measured due to high complement levels in the MSC growth media).

pathway (GO: 0030449), where 13 genes out of 44 were upregulated in MSC-NPs compared to MSCs (Fig 7). PCR validation demonstrated a significant increase in multiple complement pathway molecules including C3 (Fig 7). Although we were unable to measure C3 secretion by MSCs due to high C3 levels in MSC growth media, the level of secreted C3 from MSC-NPs significantly correlated with the level of C3 gene expression in the cells (Fig 7).

## Discussion

A critical question in the development of autologous MSC-based cell therapy for MS is whether MS disease itself impairs MSCs in any way that may affect their therapeutic efficacy. In the current study, RNA sequencing of MSCs derived from donors with progressive MS revealed negligible differences in the transcriptomic signature compared to MSCs from non-MS donors. The equivalence of MSCs from MS and non-MS donors was also shown in terms of similarity in MSC growth in culture, suppression of T cell proliferation, promotion of oligodendrocyte differentiation, and gene expression of candidate trophic and immunomodulatory genes [10]. This contrasts with other reports suggesting that MSCs derived from progressive MS patients display reduced expansion rate, premature aging, dysregulated antioxidant capacity, and impaired immunoregulatory and neuroprotective function [15–18]. The discrepancy may be due to differences in our methods of MSC isolation and *ex vivo* expansion, including isolation of MSCs from sternal bone marrow aspirates, the incubation of MSCs in low oxygen (5%), and substitution of FBS in the growth media with either autologous serum (AS) or human platelet lysate [8,10]. A notable limitation in our studies is the relatively low number of healthy donor-derived MSCs compared to other reports [15,16]. Nevertheless, our studies demonstrate that MS-derived MSCs and MSC-NPs appear to be phenotypically similar to non-MS MSCs, with initial evidence that they are therapeutically efficacious in a clinical setting [8,19,20].

MSC-NPs are derived from *ex vivo* expanded MSCs and are characterized based on their similarity to neural progenitor cells with respect to neural gene upregulation and neurosphere morphology [10]. Based on these initial characteristics, a panel of genes was selected for identity testing of MSC-NP for batch release during cGMP manufacturing, specifically the upregulation of neural genes *CXCR4*, *NES*, *TLR2*, and *HGF* and downregulation mesodermal genes *ACTA2* and *THY1* in MSC-NPs compared to the MSCs from which they are derived. Results from RNA sequencing demonstrated the enrichment of genes regulating neuronal development and differentiation in MSC-NPs, which further supports the use of neural genes in identity testing. These data will potentially identify additional genes that better confirm identity of

MSC-NP batches. Despite these data, it remains unclear how the neural identity of MSC-NPs relates to their therapeutic mechanism of action. MSC-NPs do not differentiate into neural lineages upon transplantation in EAE mice, where results suggested that their ability to improve paralysis scores in EAE was due to an indirect trophic and immunomodulatory effect [11].

The upregulation of cell signaling molecules in MSC-NPs compared to MSCs gives valuable insight into the proposed therapeutic mechanism of action of MSC-NPs in MS. Decades of research on MSCs has shown that their primary therapeutic mechanism of action is through the secretion of growth factors, cytokines, chemokines and other bioactive molecules that mediate tissue repair and immunomodulation through paracrine mechanisms [2]. Similarly, MSC-NPs appear to exhibit comparable bystander influence on oligodendrocytes, T cells, and microglia *in vitro* [10]. The enrichment of a broad range of cell signaling molecules in MSC-NPs identified by RNA sequencing suggests that MSC-NPs may have more potent paracrine signaling capacity compared to MSCs. Although few side-by-side comparison studies have been reported, a recent study in the EAE mouse model of MS demonstrated that MSC-NPs were more potent than MSCs in the attenuation of EAE [21].

ECM proteins are abundantly expressed in MSCs where they are well-known to play a key role in MSC-mediated tissue regeneration by restoring tissue structure, directing cell migration, promoting angiogenesis and stimulating nerve growth [14]. Although the upregulation of ECM proteins in MSC-NPs is likely related to their spheroid morphology, there is evidence that they also play a role in enhanced cell survival and regenerative function [22]. TIMP-1, for example, has been shown to promote oligodendrocyte differentiation and enhance CNS myelination, suggesting that MSC-NP-derived TIMP-1 may play a therapeutic role in the context of MS [23,24]. Similarly, osteopontin, which is significantly upregulated in MSC-NPs, has been shown to promote oligodendrocyte proliferation and differentiation suggesting that it is beneficial to remyelination [25]. While further validation is required, the upregulation of ECM proteins in MSC-NPs points to a potential therapeutic mechanism by which MSC-NPs may modify neuroinflammation and remyelination in MS.

The enrichment of cell signaling molecules in MSC-NPs also reveals novel candidate factors that may be associated with the paracrine effects of MSC-NPs. For example, the upregulation of *SERPINF1* gene and the associated increase in the release of the neuroprotective factor PEDF [26]. Future studies will examine the role of PEDF in MSC-NP-mediated therapeutic mechanisms in the CNS. Another candidate paracrine factor is HGF, due to its role in mediating functional recovery after MSC injection in mouse models of MS [27]. Surprisingly, although the gene expression of growth factor HGF was found to be upregulated in MSC-NPs, we did not detect a consistent increase in the release of HGF by MSC-NPs compared to MSCs. MSC-NPs were also enriched for genes regulating macrophage and glial cell migration, and oligodendrocyte differentiation. The significance of these pathways is supported by previous studies demonstrating that MSC-NPs promote oligodendrocyte differentiation *in vitro* and are associated with increased recruitment of glial progenitors to demyelinated lesions in EAE [10,11]. Furthermore, a recent study found that MSC-NPs suppress microglial activation via paracrine mechanisms [28]. The genes associated with oligodendroglial and microglial pathways that are upregulated in MSC-NPs, including *CSF1*, *CCL2*, *CCL5* and *MMP14* require further investigation into their contribution to MSC-NP therapeutic action.

An unexpected finding from this study is the enrichment of genes regulating complement activation in MSC-NPs. The physiological relevance of the complement pathway has previously been studied in MSCs, which express complement proteins C3 and C5 as well as receptors C3aR and C5aR [29,30]. Stimulation of MSCs with C3a and C5a is associated with chemotaxis, protection from oxidative damage, and modulation of the immunosuppressive function of MSCs, suggesting a potential mechanism underlying MSC migration to areas of

tissue injury and their participation in tissue repair [29–31]. Whether these functional aspects are enriched in MSC-NPs along with complement remains to be determined. However, the upregulation of complement is consistent with the upregulation of cell signaling capacity of MSC-NPs and their potential for tissue repair.

In conclusion, this study characterizes the transcriptomic shift occurring in MSC-NPs after derivation from MSCs and highlights the increased potential for cell signaling in MSC-NPs. Whether or not these gene expression differences translate into increased secretion of specific proteins that impact the potency or therapeutic efficacy of MSC-NP-based cell therapy remains to be determined.

## Supporting information

**S1 Table. All GO terms enriched in MSCs compared to MSC-NPs.** Gene ontology analysis was performed using GORilla. Target genes ("n") were all 1,467 DEGs downregulated in MSC-NPs compared to MSCs, of which 1,234 genes were associated with a GO term. Background set of genes ("N") consisted of all 24,196 genes detected by RNAseq, of which 17,705 genes were associated with a GO term. "B" is the total number of genes associated with each specific GO term, and "b" is the number of genes in the intersection. Enrichment factor = (b/n) / (B/N). Asterisks indicate pathways selected for manuscript.
(PDF)

**S2 Table. List of 133 overlapping genes in mitotic cell cycle process GO pathway (GO term GO:1903047) that were enriched in MSCs compared to MSC-NPs.**
(PDF)

**S3 Table. All GO terms enriched in MSC-NPs compared to MSCs.** Gene ontology analysis was performed using *GORilla*. Target genes ("n") were all 2,156 DEGs upregulated in MSC-NPs compared to MSCs, of which 1,799 genes were associated with a GO term. Background set of genes ("N") consisted of all 24,196 genes detected by RNAseq, of which 17,707 genes were associated with a GO term. "B" is the total number of genes associated with each specific GO term, and "b" is the number of genes in the intersection. Enrichment factor = (b/n) / (B/N). Asterisks indicate pathways selected for manuscript.
(PDF)

**S4 Table. List of 165 overlapping genes in nervous system development GO pathway (GO term GO:0051960) that were enriched in MSC-NPs.**
(PDF)

**S5 Table. List of 126 overlapping genes in cell-cell signaling GO pathway (GO term GO:0007267) that were enriched in MSC-NPs.**
(PDF)

**S6 Table. List of 84 overlapping genes in extracellular matrix organization GO pathway (GO term GO:0030198) that were enriched in MSC-NPs.**
(PDF)

## Acknowledgments

The authors would like to thank Cara Kizilbash for technical assistance.

## Author Contributions

**Conceptualization:** Violaine K. Harris, Saud A. Sadiq.

**Formal analysis:** Violaine K. Harris.

**Funding acquisition:** Saud A. Sadiq.

**Investigation:** Jaina Wollowitz, Jacelyn Greenwald, Alyssa L. Carlson.

**Project administration:** Violaine K. Harris, Saud A. Sadiq.

**Resources:** Saud A. Sadiq.

**Supervision:** Violaine K. Harris, Saud A. Sadiq.

**Validation:** Jaina Wollowitz, Jacelyn Greenwald, Alyssa L. Carlson.

**Visualization:** Violaine K. Harris, Jaina Wollowitz.

**Writing – original draft:** Violaine K. Harris.

**Writing – review & editing:** Jaina Wollowitz, Jacelyn Greenwald, Alyssa L. Carlson, Saud A. Sadiq.

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
