## [Decision Letter · Decision Letter 0]

24 Apr 2023

PONE-D-23-03957Mesenchymal stem cell-neural progenitors are enriched in cell signaling molecules implicated in their therapeutic effect in multiple sclerosis.PLOS ONE

Dear Dr. Harris,

Thank you for submitting your manuscript to PLOS ONE. After careful consideration, we feel that it has merit but does not fully meet PLOS ONE’s publication criteria as it currently stands. Therefore, we invite you to submit a revised version of the manuscript that addresses all the points raised during the review process.

Two experts have evaluated the manuscript and further amendments are suggested. The gene expression data are well analyzed and and the addition of protein data for selected genes well supports the main conclusions of the study. No further experiments are needed, but in the Discussion the limitations of the present study should be addressed.==============================

We look forward to receiving your revised manuscript.

Kind regards,

Mária A. Deli, M.D., Ph.D.

Academic Editor

PLOS ONE

Journal Requirements:

2. We note that you have a patent relating to material pertinent to this article. Please provide an amended statement of Competing Interests to declare this patent (with details including name and number), along with any other relevant declarations relating to employment, consultancy, patents, products in development or modified products etc. Please confirm that this does not alter your adherence to all PLOS ONE policies on sharing data and materials, as detailed online in our guide for authors http://journals.plos.org/plosone/s/competing-interests by including the following statement: "This does not alter our adherence to  PLOS ONE policies on sharing data and materials.” If there are restrictions on sharing of data and/or materials, please state these. Please note that we cannot proceed with consideration of your article until this information has been declared.

Reviewers' comments:

Reviewer's Responses to Questions

**Comments to the Author**

1. Is the manuscript technically sound, and do the data support the conclusions?

Reviewer #1: Yes

Reviewer #2: Yes

2. Has the statistical analysis been performed appropriately and rigorously? 

Reviewer #1: Yes

Reviewer #2: Yes

3. Have the authors made all data underlying the findings in their manuscript fully available?

Reviewer #1: Yes

Reviewer #2: Yes

4. Is the manuscript presented in an intelligible fashion and written in standard English?

Reviewer #1: Yes

Reviewer #2: Yes

5. Review Comments to the Author

Reviewer #1: The use of MSC cells in regenerative medicine is a very important and promising field. The discovery of the mechanisms behind the positive effects is particularly important when the cell products are put into clinical trials. From this point of view, the work can provide useful and interesting results in the exploration of the mechanism of MSCs and MSC-NPs. Although the number of samples is low in this study (as the authors properly point it out),

I consider it important that the samples came from the already completed clinical trial, so no further invasive intervention was necessary.

The data seem to be well processed, and the conclusions are moderate.

I would like to make only a few comments, mainly concerning the discussion part as follows:

- in Figure 1A, although I agree that there is no difference between the MS and control groups, it is not clear why the authors say that the samples are clustered into one group? Rather, it cannot be interpreted as clustering or interpret what it means in this case if they are not clustered together.

The differences between MS donors are addressed again in the discussion "Furthermore, examination of the >100 MS donor-derived MSCs manufactured for autologous administration into clinical trial subjects (cliniclatrials.gov ID NCT01933802, NCT03355365 and NCT03822858) has not revealed any correlation between donor age or disease duration and population doubling time of MSCs (data not shown).” That's a very strong claim without presenting the data, especially as this is a controversial issue in the field. I recommend rephrasing the sentence or providing supporting data.

- Due to the spheroid shape, it is completely understandable that the expression of extracellular matrix genes changes in MSC-NPs, but what relevance can this have in vivo conditions? This should also be addressed in the discussion.

Reviewer #2: The paper was described correctly and clearly. Unfortunately, with 2 control donors, it is difficult to draw conclusions. However, the main experimental gaps relate to further research. The evaluation of gene expression seems to be only the beginning. Subsequently, the level of proteins and the secretory properties of the cells should be assessed.

Changes in gene expression in differentiated cells are obvious. The question is whether they have an impact on the secretion and potential therapeutic properties of the cells.

If the authors carry out further experiments, then the paper will be very interesting.

6. PLOS authors have the option to publish the peer review history of their article (what does this mean?). If published, this will include your full peer review and any attached files.

Reviewer #1: No

Reviewer #2: No

---

## [Author Response · Author response to Decision Letter 0]

17 May 2023

Reviewer #1: The use of MSC cells in regenerative medicine is a very important and promising field. The discovery of the mechanisms behind the positive effects is particularly important when the cell products are put into clinical trials. From this point of view, the work can provide useful and interesting results in the exploration of the mechanism of MSCs and MSC-NPs. Although the number of samples is low in this study (as the authors properly point it out),

I consider it important that the samples came from the already completed clinical trial, so no further invasive intervention was necessary.

The data seem to be well processed, and the conclusions are moderate.

I would like to make only a few comments, mainly concerning the discussion part as follows:

- in Figure 1A, although I agree that there is no difference between the MS and control groups, it is not clear why the authors say that the samples are clustered into one group? Rather, it cannot be interpreted as clustering or interpret what it means in this case if they are not clustered together.

Author response: The reviewer is correct in pointing out that the term ‘clustering’ was inappropriately used in the sentence. The Results section on page 10 of the manuscript was revised to say the following “PCA analysis (Fig 1) showed minimal variance between MSCs across groups.”

The differences between MS donors are addressed again in the discussion "Furthermore, examination of the >100 MS donor-derived MSCs manufactured for autologous administration into clinical trial subjects (cliniclatrials.gov ID NCT01933802, NCT03355365 and NCT03822858) has not revealed any correlation between donor age or disease duration and population doubling time of MSCs (data not shown).” That's a very strong claim without presenting the data, especially as this is a controversial issue in the field. I recommend rephrasing the sentence or providing supporting data.

Author response: The above-mentioned sentence has been removed from the revised manuscript, as the point is sufficiently made without it. 

- Due to the spheroid shape, it is completely understandable that the expression of extracellular matrix genes changes in MSC-NPs, but what relevance can this have in vivo conditions? This should also be addressed in the discussion.

Author response: The following paragraph and references were added to the Discussion page 22:

“ECM proteins are abundantly expressed in MSCs where they are well-known to play a key role in MSC-mediated tissue regeneration by restoring tissue structure, directing cell migration, promoting angiogenesis and stimulating nerve growth [14]. Although the upregulation of ECM proteins in MSC-NPs is likely related to their spheroid morphology, there is evidence that they also play a role in enhanced cell survival and regenerative function [22]. TIMP-1, for example, has been shown to promote oligodendrocyte differentiation and enhance CNS myelination, suggesting that MSC-NP-derived TIMP-1 may play a therapeutic role in the context of MS [23, 24]. Similarly, osteopontin, which is significantly upregulated in MSC-NPs, has been shown to promote oligodendrocyte proliferation and differentiation suggesting that it is beneficial to remyelination [25]. While further validation is required, the upregulation of ECM proteins in MSC-NPs points to a potential therapeutic mechanism by which MSC-NPs may modify neuroinflammation and remyelination in MS.”

Reviewer #2: The paper was described correctly and clearly. Unfortunately, with 2 control donors, it is difficult to draw conclusions. However, the main experimental gaps relate to further research. The evaluation of gene expression seems to be only the beginning. Subsequently, the level of proteins and the secretory properties of the cells should be assessed.

Changes in gene expression in differentiated cells are obvious. The question is whether they have an impact on the secretion and potential therapeutic properties of the cells.

If the authors carry out further experiments, then the paper will be very interesting.

Author response: We have confirmed increased protein expression/secretion for several of the genes described in manuscript, including PEDF and HGF in Figure 4, and TIMP-1 and Osteopontin in Figure 6. Additional validation of protein secretion and the contribution of these proteins to the therapeutic properties of MSC-NPs, while outside the scope of this particular study, is an important future direction of this research. We revised the last sentence in the manuscript discussion to highlight this point. “Whether or not these gene expression differences translate into increased secretion of specific proteins that impact the potency or therapeutic efficacy of MSC-NP-based cell therapy remains to be determined.”

---

## [Decision Letter · Decision Letter 1]

2 Aug 2023

Mesenchymal stem cell-neural progenitors are enriched in cell signaling molecules implicated in their therapeutic effect in multiple sclerosis.

PONE-D-23-03957R1

Dear Dr. Harris,

We’re pleased to inform you that your manuscript has been judged scientifically suitable for publication and will be formally accepted for publication once it meets all outstanding technical requirements.

Kind regards,

Mária A. Deli, M.D., Ph.D.

Academic Editor

PLOS ONE

Additional Editor Comments (optional):

Reviewers' comments:

Reviewer's Responses to Questions

**Comments to the Author**

1. If the authors have adequately addressed your comments raised in a previous round of review and you feel that this manuscript is now acceptable for publication, you may indicate that here to bypass the “Comments to the Author” section, enter your conflict of interest statement in the “Confidential to Editor” section, and submit your "Accept" recommendation.

Reviewer #1: All comments have been addressed

2. Is the manuscript technically sound, and do the data support the conclusions?

Reviewer #1: Yes

3. Has the statistical analysis been performed appropriately and rigorously? 

Reviewer #1: Yes

4. Have the authors made all data underlying the findings in their manuscript fully available?

Reviewer #1: Yes

5. Is the manuscript presented in an intelligible fashion and written in standard English?

Reviewer #1: Yes

6. Review Comments to the Author

Reviewer #1: In my opinion, the manuscript improved during the review process and the authors gave satisfactory answers to all my questions.

7. PLOS authors have the option to publish the peer review history of their article (what does this mean?). If published, this will include your full peer review and any attached files.

Reviewer #1: No

---

## [Editor Report · Acceptance letter]

4 Aug 2023

PONE-D-23-03957R1 

Mesenchymal stem cell-neural progenitors are enriched in cell signaling molecules implicated in their therapeutic effect in multiple sclerosis. 

Dear Dr. Harris:

I'm pleased to inform you that your manuscript has been deemed suitable for publication in PLOS ONE. Congratulations! Your manuscript is now with our production department. 

Kind regards, 

on behalf of

Prof. Mária A. Deli 

Academic Editor

PLOS ONE